# The Relationship between Maternal Functioning and Mental Health after Childbirth in Iranian Women

**DOI:** 10.3390/ijerph17051558

**Published:** 2020-02-28

**Authors:** Sevda Gholizadeh Shamasbi, Jennifer L. Barkin, Solmaz Ghanbari-Homayi, Ommlbanin Eyvazzadeh, Mojgan Mirghafourvand

**Affiliations:** 1Department of Midwifery, Faculty of Nursing and Midwifery, Tabriz University of Medical Sciences, Tabriz 977-5138947, Iran; s.gholizadeh7007@gmail.com (S.G.S.); ommolbanineyvazzadeh@gmail.com (O.E.); 2Department of Community Medicine, Mercer University School of Medicine, Macon, GA 31207, USA; barkin_jl@mercer.edu; 3PhD of Midwifery, Students’ Research Committee, Tabriz University of Medical Sciences, Tabriz 977-5138947, Iran; homayisolmaz@gmail.com; 4Midwifery Department, Social Determinants of Health Research Center, Tabriz University of Medical Sciences, Tabriz 977-5138947, Iran

**Keywords:** maternal functioning, mental health, postpartum

## Abstract

The postpartum period is critical for new mothers, in terms of performing maternal functions, which can be affected by physical or psychological complications. The purpose of the present study is to determine the relationship between maternal functioning and mental health in the postpartum period. This cross-sectional descriptive-analytic study was conducted on 530 eligible women who referred to health centers in Tabriz, Iran in 2018. The participants were selected through randomized cluster sampling, and data were collected by using a socio-demographic characteristics questionnaire, Mental Health Inventory (MHI), and the Barkin Index of Maternal Functioning (BIMF). These assessments were collected between 1 and 4 months postpartum. The relationship between maternal functioning and mental health was determined by conducting bivariate analysis via Pearson and Spearman correlation analysis and the general linear model (GLM) in a multivariate analysis. The mean (SD) mental health score in women was 79.1 (15.0) in the obtainable score range of 18 to 108, and the mean (SD) BIMF score in women was 97.4 (12.9) in the obtainable score range of 0 to 120. Based on Pearson or Spearman correlations, mental health and its sub-domains had positive, significant correlations with infant care, mother–child interaction, mental well-being, social support, management, adjustment, self-care, and maternal functioning (*p* < 0.001). Based on the GLM, increased maternal functioning was associated with higher total mental health score, having a moderate income, and receiving support for infant care (*p* < 0.05). High levels of postpartum mental health can have a positive impact on maternal functioning. Additionally, having support with infant care tasks can also improve functioning.

## 1. Introduction

Maternal functioning is an important variable in infant care during the first 12 months of birth [1] and throughout the life course of parenthood. Mothers feel largely responsible for performing infant-related tasks such as feeding, caring, changing clothes, and clinical visits [2]. Maternal functioning during the postpartum period is a multidimensional concept that includes personal care, infant and family care, and social and occupational activities. Most mothers often require at least 6 months for complete adjustment to new motherhood [2,3]. Factors that may affect maternal functioning during the postpartum period in the general population include parity, labor experience, delivery type [4], receiving or not receiving social support [5], postpartum psychological status [6], and maternal and neonatal complications [4]. Postpartum maternal functioning and infant-related tasks act as important factors for the infant’s growth and development in the first year of life [1,2,7,8,9,10]. In addition, mothers must integrate their new responsibilities related to infant care into their existing set of duties/activities; reprioritization is often a necessity (and healthy) for new mothers. Although new motherhood can be intensely satisfying, it can also trigger mental health challenges [2].

Mental health is defined as, “a state of well-being in which every individual realizes his or her own potential, can cope with the normal stresses of life, can work productively and fruitfully, and is able to make a contribution to her or his community” [11,12,13,14]. The findings of a study have indicated that the most common problem in new mothers during postpartum is baby blues, followed by postpartum depression and anxiety disorders [15]. Maternal mental health problems during the prenatal period are commonly studied in high-income countries, whereas little research has been conducted in low- and middle-income countries [16]. Major health problems during prenatal and postnatal periods occur in 1 out of every 3–5 women in developing countries and in 1 out of every 10 women in developed countries [17].

The vulnerability of women increases in the postpartum period [18] in terms of mental health and can impact overall maternal functioning [7] through changing the sense of self-efficacy [19,20] and maternal gratification [21,22,23]. Maternal functioning plays a key role in the development, health, and support of the child; therefore, in the postpartum period, maternal functioning may be impaired and affected by maternal emotions and feelings. Evidence suggests that the quality of care given to the infant in the first year of life by mothers suffering from impaired mental health is reduced, resulting in decreased health of the infant in the future [2]. Aktan et al. [24] and Mcveigh et al. [25,26] showed an inverse relationship between anxiety and maternal functioning. Barkin et al. [27], Posmontier et al. [7], and Fathi et al. [28] in Iran revealed an inverse relationship between depression and maternal functioning. It should be noted that Iranian and non-Iranian studies did not analyze the relationship between maternal functioning and health aspects other than anxiety and depression. In addition, the postpartum period is an opportune time to monitor maternal physical and mental health, starting and continuation of breastfeeding, infants’ physical health, and caring for both mother and infant according to their needs. Moreover, it is possible that maternal health problems can persist throughout the first postpartum year [18,29,30]. Therefore, in order to prevent unnecessary complications that may require treatment, assessment of functional status of new mothers is essential. Evaluation of maternal functioning may lend itself best to skill-building interventions—rather than traditional therapeutic or pharmacological treatment for depression/anxiety [31]. The aim of the present study is to determine the relationship between maternal functioning and mental health (according to the Mental Health Inventory) during the postpartum period. The authors hypothesize that maternal functioning (as measured by the Barkin Index of Maternal Functioning (BIMF) [27]) will have a direct, significant relationship with mental health scores (as measured by the Mental Health Inventory (MHI) [32]).

## 2. Methods

### 2.1. Study Design and Participants

This cross-sectional study was conducted on 530 women visiting the health centers of Tabriz, Iran. The inclusion criteria were: mothers with first child, pregnancy with only one child, normal pregnancy and labor, self-reported physical health, and willingness to participate in the study. Exclusion criteria were a history of mental illness before pregnancy, during pregnancy, or after delivery (according to the mother), stressful incidents such as divorce, death of family members, diagnosis of an incurable illness in a family member during the three previous months, newborn admission to the neonatal ward of hospital due to illness, newborn abnormalities (these events can lead to mental disorders), and participant withdrawal from the study during completion of the questionnaire.

Considering the standard deviation (SD) = 17, precision (*d*) = 0.02 around the mean (*m* = 80), and an α = 0.05, the sample required 434 participants; however, 530 people were recruited and analyzed in this study due to possible attrition [33].

### 2.2. Sampling

After the approval of the project, obtaining the ethical code from the Ethics Committee of the Research Deputy of Tabriz University of Medical Sciences (Ethics code: IR.TBZMED.REC.1397.147), the participants were selected through cluster sampling. First, one fourth of the health centers of Tabriz were selected randomly through the website www.random.org. Subsequent to this initial step, the list of mothers giving birth one to four months ago was provided to the research team according to the health records in each center. After that, the required number of samples for each center or base was determined through the proportional to size sampling method and randomly selected from the list. Using the phone number in each file, the research coordinator called the mothers, provided a thorough description of the study, and invited them to participate in the research. Five hundred and thirty women giving birth one to four months ago were selected from health centers, based on the number of samples assigned to each center. At the in-person session, the research objectives were described fully to mothers, and their eligibility criteria were checked. The eligible women were ensured of information confidentiality, and informed consent was obtained. Lastly, the research questionnaires were filled out by the participants.

### 2.3. Data Collection Instruments

#### 2.3.1. Socio-demographics

A researcher-developed socio-demographic questionnaire was used in this study. It included items pertaining to maternal age, education level, employment, wanted or unwanted pregnancy. Other variables, deemed to be relevant by the research team, were also included and can be viewed in Table 1. The validity of this questionnaire was verified for content and face validity. Specifically, the questionnaire was provided to 10 faculty members with appropriate expertise for review, and the necessary amendments were made.

#### 2.3.2. Maternal Functioning

Information about maternal functioning was collected using the BIMF. This 20-item tool includes the subjective areas of self-care, infant care, mother–child interaction, psychological well-being, social support, management, and adjustment to assess postpartum functioning in new mothers. The BIMF employs a Likert-style scale [27] and reponse options include strongly disagree (score 0), disagree (score 1), somewhat disagree (score 2), neutral (score 3), somewhat agree (score 4), agree (score 5), and strongly agree (score 6). The participants were asked to give the best answer regarding their feelings over the past two weeks. The mean total score of maternal functioning can range from 0 to 120. Higher total scores indicate higher functioning levels. In this study, Cronbach’s alpha coefficient was 0.88, indicating good internal consistency, and the intra correlation coefficient (ICC) was equal to 0.85.

#### 2.3.3. Mental Health

Mental health was assessed by using the Mental Health Inventory (MHI) developed by Veit and Ware (1983). MHI has a long form (38 items) [34] and a short form (18 items) [32]. The reason for using MHI in this study rather than other questionnaires such as the General Health Questionnaire (GHQ) [35] or the Edinburgh Postnatal Depression scale (EPDS) [36] was that MHI was designed for normal populations, whereas GHQ is essentially a diagnostic tool developed for clinical populations. Additionally, only depression is assessed by the EPDS while MHI sub-domains include anxiety, depression, behavioral control, and positive mood. The MHI short form was utilized in this study. The 18-item version is a mental health screening tool for adults in two areas of general health and psychological distress. Options include: “all of the time” (score 1), “most of the time” (2), “a good bit of the time” (3), “some of the time” (4), “a little bit of the time” (5), and “none of the time” (6). The psychometrics study of the MHI was conducted by Meybodi in Iran in 2011 and indicated that this scale is a reliable and valid tool for using in Iranian population [37].

### 2.4. Statistical Analysis

Data were entered into SPSS version 21 (IBM Corporation, Chicago, IL, USA). After determining the normality of the distribution by examining skewness and kurtosis tests, the data were analyzed by using descriptive statistics. Frequency, percentage, mean, and standard deviation were calculated for all socio-demographic characteristics, and for maternal functioning and mental health. Pearson and Spearman correlation tests were used to determine the relationship between maternal functioning and its dimensions and mental health. An independent t-test and one-way ANOVA were used to determine the relationship between socio-demographic characteristics and maternal functioning. All variables with p < 0.05 based on bivariate tests (Pearson and Spearman correlation tests, independent t-test, and one-way ANOVA) were entered into a general linear model (GLM) for the purposes of multivariate analysis.

## 3. Results

A total of 530 women participated in the study from July 1, 2018 to January 10, 2019 and completed the study instruments. In terms of socio-demographic characteristics, the mean (SD) birth weight of the infants was 3129.6 (511.7) grams, the gestational age at the birth was 38.5 (2.1) weeks, and the mean age of mothers was 27.0 (5.4) years. Table 1 shows the other socio-demographic characteristics of the participants.

The mean (SD) mental health score in women was 79.1 (15.0) in the obtainable score range of 18 to 108. The mean (SD) of anxiety, depression, behavioral control, and positive mood sub-domains was 21.7 (4.7) [obtainable score range = 5 to 30], 18.8 (4.1) [4 to 24], 17.0 (3.0) [4 to 24], and 17.1 (4.2) [4 to 24] (Table 2).

The mean (SD) maternal functioning score of mothers was 97.4 (12.9) in the obtainable score range of 0 to 120. The mean (SD) of self-care, infant care, mother–child interaction, psychological well-being, social support, management, and adjustment sub-domains was 14.4 (3.1) [obtainable score range = 0 to 18], 11.2 (1.2) [0 to 12], 15 (2.4) [0 to 18], 47.3 (6.8) [0 to 60], 13.6 (3.8) [0 to 18], 27.7 (5.0) [0 to 36], and 10.6 (1.5) [0 to 12] (Table 2).

Mental health and its sub-domains had a positive, significant correlation with the total score of maternal functioning and the infant care, mother–child interaction, psychological well-being, social support, management, adjustment, and self-care sub-domains (*p* < 0.001) (Table 3).

The results of an independent *t*-test and one-way ANOVA showed that maternal functioning had a positive, significant relationship with maternal age, spouse’s age, income adequacy, interest in newborn sex, life satisfaction, and receiving support for infant care (*p* < 0.05) (Table 1). These variables, along with the mental health variable, were entered into the GLM as independent variables. The dependent variable was maternal functioning. The variables of total score of mental health, mother’s age and husband’s age were quantitative variables and entered into the model as covariate and the variables of income adequacy, interest in newborn sex, life satisfaction, and receiving support for infant care were qualitative and categorical that entered into the model as fixed factors (fixed factors are categorical independent variables). The results of GLM demonstrated that receiving help for infant care, moderate levels of income, and satisfactory mental health had the relationship with maternal functioning (*p* < 0.05) and predicted 39.6% of the variance in maternal functioning. The variables of mother’s age, husband’s age, interest in sex of newborn, and satisfaction of life had no relationship with maternal functioning (Table 4).

## 4. Discussion

In this study, mothers had a relatively high level of maternal functioning and mental health score. Among the maternal functioning sub-domains, the highest score pertained to psychological well-being, whereas the lowest score was related to adjustment. Maternal functioning and its sub-domains had direct, significant relationships with mental health and its sub-domains. Increased maternal functioning was associated with a higher total mental health score, having a moderate income, and receiving support for infant care.

In this study, the results indicated that mothers had reasonably high maternal functioning (mean score = 97.4). In a study conducted by Williams (2018) et al. (2018) [38] on 46 mothers of infants admitted to a Level III NICU in the midwestern United States, the mean score of BIMF was 96.1. A study conducted by Barkin et al. (2017) [39] on 128 women receiving postpartum obstetrical care at a large medical center in medically underserved, middle Georgia, resulted in an average BIMF total score of 104. In other studies carried out by Barkin et al. (2014) [27,33,40] on depressed women, mean BIMF score was lower (in the 80s). The results of these studies, as a whole, are intuitive given that the depressed subpopulation(s) of women had the lowest functioning scores. The high score of maternal functioning shows that women were satisfied and were adjusting to the maternal role sufficiently.

In the present study, among all maternal functioning sub-domains, the highest scores pertained to psychological well-being and the lowest score to adjustment. In a study on 305 Iranian women at 4 months postpartum, the highest mean score of functional status belonged to infant care sub-domain and the lowest mean score was for social and community activities [41]. In Soltanpoor’s et al. (YEAR) study on 165 women, the mean score of infant care and self-care was high, and the mean score of social activity was low [42]. Both of these studies used the Inventory of Functional Status after Childbirth (IFSAC), which has different dimensions and a different scoring range than the tool used in the present study. In addition, adequate time to rest and practice self-care can lead to the quick recovery of mothers in the postpartum period and maternal functioning improvement [43].

In the present study, the maternal functioning sub-domain scores increased with an increase in the mental health total and sub-domain scores. According to Barkin et al. (2016) in the United States, maternal functioning had a significant inverse relationship with depression, atypical depression, and bipolar diagnosis [33]. In a study by Aktan et al. (2010) [24] in the United States, postpartum anxiety had an inverse relationship with maternal functioning and the sub-domains of self-care and household chores, using the IFSAC. These studies are consistent with the present research. In a prospective study conducted by McVeigh et al. (2000) in Australia, using the IFSAC, maternal functional status and its sub-domains of self-care and social activity had inverse, significant relationships with anxiety [26]. According to a cross-sectional study carried out by Posmontier et al. (2008) in the United States, women with symptoms of depression were less likely to excel at social functions, household chores, personal care, and self-care. However, depression had no effect on breastfeeding [7]. The results of these studies are also consistent with the present research. Several factors such as anxiety and depression (mental health disorders) can affect maternal recovery and maternal functioning in the postpartum period. Anxiety and depression during the postpartum period can lead to the impaired adjustment of mothers, delay their recovery, and even impede the actions that are necessary to maintain their health [31,43].

The results of this study indicated that receiving support for infant care can lead to enhanced maternal functioning. Receiving social support is a powerful predictor for postpartum maternal health and functioning [44]. In a study conducted by Aktan et al. (2010) in the United States, receiving support was directly associated with the self-care and social activities sub-domains of functional status [24]. Some mothers believed that partners, friends, or even family members could help mothers with infant care and could even provide mothers with opportunities to socialize and time for leisure, self-care, and rest. Support from others may afford new mothers the time and resources to tend to themselves physically and emotionally [45]. A study by Brown et al. (1998) in Australia indicated that mothers wanted help with infant care because the fatigue and health problems that they experienced after a difficult vaginal delivery did not allow them to properly care for their infants [46]. In an Australian study by McVeigh et al. (1995), satisfaction with support from family and friends related to infant care had a positive impact on maternal functioning, as measured by the IFSAC [26]. According to mothers in a study by Jirapaet et al. (2001) in Thailand, social support had a positive effect on maternal functioning [47]. In a study by Herba et al. (2016) in low- and middle-income countries, the relationship between mental health of mothers and child outcomes depends on receiving support [44]. Considering the fact that receiving support is a predictor of maternal functioning, interventions of social support should be included in programs for improvement of maternal functioning.

In this particular study, having a low to moderate income was associated with higher maternal functioning. However, this result is not in agreement with the majority of the scientific literature which indicates that increased resources are protective [28,43]. In a cross-sectional study conducted by Fathi et al. (2018) in Iran, a direct significant relationship existed between income and maternal functional status [28]. Additionally, in a study carried out by Ahn et al. (2007) in the United States, higher maternal functioning at 6 weeks postpartum had a strong relationship with high family income [43]; this result is intuitive, as having financial resources generally equates having greater range of choices. Postpartum recovery is easier and quicker for people with low socio-economic status who have no access to these facilities because they cannot afford them [48]. According to Mangham et al., low- and middle-income countries have added routine public health services for infant care, breastfeeding practices in hospitals, and the use of neonatal care standards in their routine programs to improve neonatal care and maternal function [49]. In Iran, as a middle-income country, all above mentioned services are provided for mothers in health centers for free. By visiting public health centers, mothers in low-income families can receive all services and this may lead to better maternal functioning in this social class.

The large sample size (*n* = 530) was a strength of this study. Regarding the research limitations, this was a cross-sectional study in which the cause-and-effect relationships cannot be demonstrated. Therefore, the relationship between maternal functioning and mental health demonstrated in this study does not necessarily indicate cause and effect. Thus, future prospective, longitudinal researches are needed in this regard. This study was also conducted on primiparous and singleton women, which does not represent the circumstances of all new mothers. Therefore, this study cannot be generalized to multiparous women and multiple pregnancies because the presence of children at home and multiple pregnancies can impact maternal functioning. A future study should be conducted regarding the association between maternal functioning and mental health in multiparous women. Additionally, the results of this study are based only on self-report measures, thus, observational measures should be considered in the future studies. Since the mothers who had experienced recent stressful events in their lives and/or had a history of mental disorder were excluded from this study, therefore, the findings of this study do not represent the circumstances of all new mothers.

The findings of the present study can serve as a guide for future studies and can help professionals working in various maternal health centers. Researchers are recommended to conduct studies with strong design such as cohort or clinical trials, to improve maternal functioning.

## 5. Conclusions

This study demonstrated that in our sample of Iranian women, there was a relationship between maternal mental health and overall maternal functioning and all subdomains. In addition, receiving help for infant care may lead to improved maternal functioning. This work requires confirmation in other subgroups of women including multiparous women. Due to the limitation of cross-sectional studies cause-and-effect relationships, the conducting of longitudinal, prospective studies are suggested to absolute conclusion about the relationship between maternal functioning with mental health and some socio-demographic factors such as income and receiving help for infant care. Based on the results of this types studies, screening of mothers for mental health during the postpartum period will be recommended for early diagnosis and treatment of mental disorders and improving of maternal functioning. Enhanced social support is also recommended during pregnancy and in the postpartum as it is a known, protective factor.

## Figures and Tables

**Table 1 ijerph-17-01558-t001:** The Relationship between socio-demographic and obstetrics characteristics with maternal functioning in women referred to health centers of Tabriz, Iran, 2019 (*n* = 530).

Variable	Number (Percent)	Mean (SD)	*p*-Value	Variable	Number (Percent)	Mean (SD)	*p*-Value
**Weight of newborn**	530 (100)	3129.6 (511.7)	0.701 ^‡^	**House Status**			
**Gestational age**	530 (100)	38.5 (2.1)	0.280 ^‡^	Personal	188 (35.5)	97.4 (13.9)	0.385 ^†^
**Mother’s age**	530 (100)	27.0 (5.4)	<0.001 ^‡^	Hire	213 (40.2)	96.3 (12.5)	
**Husband’s age**	530 (100)	31.9 (5.2)	0.003 ^‡^	Parent’s House	11 (2.1)	97.0 (10.8)	
**Mother’s Job**				Mother-in-law’s House	116 (21.9)	99.4 (12.2)	
House wife	504 (95.1)	97.3 (13.0)	0.447 *	Other	2 (0.4)	95.5 (12.9)	
Employed	21 (4.0)	99.5 (11.6)		**Live with**			
**Husband’s Job**				Husband	406 (76.6)	97.2 (13.1)	0.372 ^†^
Unemployed	10 (1.9)	98.1 (9.9)	0.241 ^†^	Alone	4 (0.8)	88.0 (8.7)	
Worker	152 (28.7)	98.4 (14.0)		My Family	10 (1.9)	95.7 (9.3)	
Employee	66 (12.5)	97.8 (13.2)		Husband’s family	110 (20.8)	98.5 (12.6)	
Self-employee	289 (54.5)	96.5 (12.4)		**Sex of Newborn**			
Other	13 (2.5)	103.7 (9.5)		Girl	274 (51.7)	97.1 (12.7)	0.544 *
**Mother’s Education**				Boy	256 (48.3)	97.8 (13.2)	
Primary school	67 (12.6)	94.8 (14.2)	0.222 ^†^	**Wanted Pregnancy**			
Secondary school	97 (18.3)	96.1 (13.4)		Yes	403 (76.0)	97.7 (12.8)	0.398 *
High school	67 (12.6)	97.3 (13.8)		No	127 (24.0)	96.6 (13.3)	
Diploma	195 (36.8)	98.3 (12.6)		**Interest in Sex of Newborn**			
University	104 (19.6)	98.7 (11.4)		No	20 (3.8)	92.2 (11.6)	0.018 *
**Husband** **’s Education**				Both me and my Husband	493 (93.0)	97.9 (13.0)	
Primary school	64 (12.1)	96.1 (13.4)	0.235 ^†^	Only me	10 (1.9)	89.0 (9.3)	
Secondary school	120 (22.6)	95.8 (13.3)		Only my husband	7 (1.3)	90.7 (10.0)	
High school	56 (10.6)	96.4 (12.9)		**Helping you with Baby Care**			
Diploma	192 (36.2)	99.0 (12.3)		Yes	335 (63.2)	99.9 (12.7)	<0.001 *
University	98 (18.5)	97.4 (12.9)		No	195 (36.8)	93.2 (12.1)	
**Satisfaction of Life**				**Who Help You**			
Much	270 (50.9)	102.1 (12.3)	<0.001 ^†^	Husband	114 (21.5)	100.8 (11.9)	0.101 ^†^
Moderate	253 (47.7)	92.5 (11.8)		Mother	111 (20.9)	97.3 (13.2)	
Low	7 (1.3)	94.7 (11.3)		Mother-in-low	73 (13.8)	102.3 (11.9)	
**Sufficiency of Income**				Sister	9 (1.7)	98.0 (22.0)	
Completely sufficient	57 (10.8)	103.4 (12.4)	0.001 ^†^	Sister-in-low	3 (0.6)	108.0 (9.6)	
Somewhat sufficient	393 (74.2)	96.4 (12.6)		Other	25 (4.7)	99.6 (12.2)	
Insufficient	80 (15.1)	98.3 (13.8)					

* Independent t test; ^†^ one way ANOVA; ^‡^ Pearson correlation test.

**Table 2 ijerph-17-01558-t002:** The status of maternal functioning and mental health in women referred to health centers of Tabriz, Iran, 2019 (*n* = 530).

Variable	Mean (SD)	Obtained Range	Obtainable Range
**Mental Health**	79.1 (15.0)	28–108	18–108
Anxiety	21.7 (4.7)	6–30	5–30
Depression	18.8 (4.1)	4–24	4–24
Behavior Control	17.0 (3.0)	5–24	4–24
Positive Mood	17.1 (4.2)	5–24	4–24
**Maternal Functioning**	97.4 (12.9)	41–120	0–120
Self-Care	14.4 (3.1)	2–18	0–18
Newborn Care	11.2 (1.2)	4–12	0–12
Mother–Child Interaction	15.0 (2.4)	7–18	0–18
Maternal Psychological	47.3 (6.8)	19–60	0–60
Social Support	13.6 (3.8)	3–18	0–18
Management	27.7 (5.0)	8–36	0–36
Adjustment	10.6 (1.5)	2–12	0–12

In all variables, higher scores are desirable and represent better maternal functioning or mental health.

**Table 3 ijerph-17-01558-t003:** The relationship between maternal functioning and mental health in women referred to health centers of Tabriz, Iran, 2019 (*n* = 530).

Variable	Total Score of Mental Health		Anxiety		Depression		Behavior Control		Positive Mood	
	r	*p*	r	*p*	r	*p*	r	*p*	r	*p*
**Newborn Care**	0.383	<0.001	0.305	<0.001	0.377	<0.001	0.382	<0.001	0.317	<0.001
**Mother–Child Interaction**	0.443	<0.001	0.369	<0.001	0.360	<0.001	0.387	<0.001	0.435	<0.001
**Maternal Psychological**	0.579	<0.001	0.515	<0.001	0.543	<0.001	0.504	<0.001	0.492	<0.001
**Social Support**	0.300	<0.001	0.238	<0.001	0.187	<0.001	0.309	<0.001	0.318	<0.001
**Management**	0.506	<0.001	0.456	<0.001	0.458	<0.001	0.416	<0.001	0.454	<0.001
**Adjustment**	0.415	<0.001	0.344	<0.001	0.412	<0.001	0.350	<0.001	0.365	<0.001
**Maternal Functioning**	0.571	<0.001	0.489	<0.001	0.498	<0.001	0.512	<0.001	0.519	<0.001
**Self-Care**	0.523	<0.001	0.431	<0.001	0.479	<0.001	0.436	<0.001	0.496	<0.001

**Table 4 ijerph-17-01558-t004:** The predictors of maternal functioning in women referred to health centers of Tabriz, Iran, 2019 (*n* = 530).

Variable	Β (CI 95%) *	*p*-Value
**Total score of Mental Health**	0.40 (0.3 to 0.4)	< 0.001
**Mother’s age**	−0.09 (−0.3 to 0.1)	0.463
**Husband’s age**	−0.00 (−0.2 to 0.2)	0.974
**Sufficiency of income**		
Completely sufficient	0	
Somewhat sufficient	−2.74 (−5.2 to 0.2)	0.031
Insufficient	−0.39 (−3.9 to 3.1)	0.827
**Interest in sex of newborn**		
Only my husband (Reference)	0	
Only me	−0.78 (−10.5 to 9.0)	0.875
Both me and my husband	4.13 (−3.4 to 11.7)	0.284
None	−0.67 (−9.3 to 8.0)	0.879
**Satisfaction of life**		
Much (Reference)	0	
Moderate	−6.57 (−14.3 to 1.2)	0.097
Low	−2.24 (−10.1 to 5.6)	0.576
**Helping you with baby care**		
No (Reference)	0	
Yes	4.58 (2.7 to 6.4)	<0.001
Adjusted R^2^ = 39.6%		

* Confidence interval.

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
