# Peer review of "The Relationship between Maternal Functioning and Mental Health after Childbirth in Iranian Women"

_ijerph, 2020, doi:10.3390/ijerph17051558_

Round 1
Reviewer 1 Report
This paper explores the relationship between “maternal functioning” and postpartum mental health. Although a worthy topic, the paper uses concerningly stigmatizing terminology and framing and should be rewritten to avoid stigmatizing new mothers struggling with the adjustment to motherhood or defining mothers through their symptoms. Additionally, several aspects of the paper need revision in order for it to be ready for publication, including the following:
Introduction
The term “maternal functioning” comes across stigmatizing. While it may be the most technical term, it is also potentially pejorative (i.e., a low score on the index suggests someone is a dysfunctional mother or worse broken). I would encourage the authors to use a different term such as “maternal adjustment,” which is a term the authors themselves use to describe functioning (line 41). Line 50-54: this level of detail in defining mental health seems excessive and unnecessary. More useful would be to list common mental health issues that new mothers face. Line 64: Terms like “mental health impairment” are also very stigmatizing. Please use more sensitive language such as “mothers suffering from impaired mental health.” The introduction ends with a clear statement of aim. But it also needs a list of a priori hypotheses based on the literature that is reviewed here. Or if the tests were not determined a priori, the authors need to justify why the state of the literature called for an entirely exploratory study (which it doesn’t seem to given all the literature cited). What tests did the authors plan to run and what did they expect to find based on the literature they cover.
Methods
As this is not a pregnancy/maternal health journal specifically, the authors should avoid specific jargon like “primiparity” and “singleton” and instead use descriptive language like “pregnant with first child” and “pregnant with only one child.”
Results
Table 1: I believe the variable should be “interest in SEX of newborn” not sexuality? Do the authors really mean that they assessed whether parents were interested in the sexual preferences of the newborn? There is far too much text that reiterates statistics that are available in tables. Instead, the authors should use the text to summarize the tables and characterize the sample and findings, rather than repeat figures in the tables. Line 167: information on how to interpret higher scores belongs either in the methods or as a footnote on the table. What do the authors mean by “entered into a general linear model?” Most of these results fall under the GLM. When talking about “prediction,” this should be more specifically a regression framework, which especially makes sense given that there is some temporality to some of these data. But there needs to be much more information here. What variables were controlled? For instance, in Lines 184-186, were these things predictors over and above all over predictors? Taking into account each other? Table 4 is unclear on whether this was all one model or separate models and whether dummy variables were constructed and all in one model?
Discussion:
This discussion spends far too much time summarizing results and not enough space synthesizing and discussing implications. The discussion uses too much causal language and needs to be tempered quite a bit to more accurately reflect the cross-sectional nature of this work. It too strong to state things like “higher levels of maternal mental health positively impact overall functional status…” Similarly, this evidence is not strong enough to support some of the suggestions the authors. This work is an important step, but future prospective, longitudinal research is also needed. This needs to be stated and the suggestions need to be framed in light of the need for more research. The conclusion needs expansion and tempering. What are potential next steps? What further research is need to then be able to inform on screening and prevention. Overall, though, it’s hard to comment in detail on the discussion as the results are so unclear. Once the results can be clarified, it will be easier to determine whether this discussion is appropriate.
Author Response
Response to reviewers
We thank you for their highly insightful comments which have enabled us to greatly improve the quality of our manuscript. Below, you will find our point-by-point responses to each of the comments.
Reviewer 1
Comments and Suggestions for Authors
This paper explores the relationship between “maternal functioning” and postpartum mental health. Although a worthy topic, the paper uses concerningly stigmatizing terminology and framing and should be rewritten to avoid stigmatizing new mothers struggling with the adjustment to motherhood or defining mothers through their symptoms. Additionally, several aspects of the paper need revision in order for it to be ready for publication, including the following:
We appreciate your observations regarding the term “maternal functioning.” The measure was developed for the express purpose of avoiding the stigmatization of women who did not prefer depression/anxiety treatment or evaluation. The developers made the observation that most women presenting for treatment expressed the desire to “function better” in daily life and many preferred a skill-building approach to traditional therapies.
Introduction
- The term “maternal functioning” comes across stigmatizing. While it may be the most technical term, it is also potentially pejorative (i.e., a low score on the index suggests someone is a dysfunctional mother or worse broken). I would encourage the authors to use a different term such as “maternal adjustment,” which is a term the authors themselves use to describe functioning (line 41).
We appreciate your sensitivity to this language and how it may or may not be perceived by new mothers, who are already coping with a complex adjustment. The need for a measure of maternal functioning was initially identified by a lead psychiatrist in perinatal mental health who aptly observed that most women presenting for treatment preferred to “function better” in daily life. This line of research and assessment allows for treatment akin to a skill-building approach – rather than traditional therapeutic or pharmacological treatment for depression/anxiety. We used the word “functioning” in order to remain consistent with the naming convention of the tool (“the Barkin Index of Maternal Functioning”). To address your point, we have added some language into the text explaining why this approach may very well be less stigmatizing than traditional depression evaluation.
- Line 50-54: this level of detail in defining mental health seems excessive and unnecessary. More useful would be to list common mental health issues that new mothers face.
We have summarized the definition of mental health. We have listed the common mental health issues that new mothers faces.
- Line 64: Terms like “mental health impairment” are also very stigmatizing. Please use more sensitive language such as “mothers suffering from impaired mental health.”
We have revised it.
- The introduction ends with a clear statement of aim. But it also needs a list of a priori hypotheses based on the literature that is reviewed here. Or if the tests were not determined a priori, the authors need to justify why the state of the literature called for an entirely exploratory study (which it doesn’t seem to given all the literature cited). What tests did the authors plan to run and what did they expect to find based on the literature they cover.
Thank you for pointing this out. We have added language to the aims section, restating the aim as a hypothesis. The authors have been advised to refer to specific statistical procedures in the methods section of the manuscript.
Methods
- As this is not a pregnancy/maternal health journal specifically, the authors should avoid specific jargon like “primiparity” and “singleton” and instead use descriptive language like “pregnant with first child” and “pregnant with only one child.”
We have replaced “primiparity” and “singleton” with “mothers with first child” and “pregnant with only one child.”
Results
- Table 1: I believe the variable should be “interest in SEX of newborn” not sexuality? Do the authors really mean that they assessed whether parents were interested in the sexual preferences of the newborn?
We have replaced sexuality with sex.
- There is far too much text that reiterates statistics that are available in tables. Instead, the authors should use the text to summarize the tables and characterize the sample and findings, rather than repeat figures in the tables.
We have summarized the tables in the text.
Line 167: information on how to interpret higher scores belongs either in the methods or as a footnote on the table.
We have reported this sentence at the footnote of the Table 2.
What do the authors mean by “entered into a general linear model?” Most of these results fall under the GLM. When talking about “prediction,” this should be more specifically a regression framework, which especially makes sense given that there is some temporality to some of these data. But there needs to be much more information here. What variables were controlled? For instance, in Lines 184-186, were these things predictors over and above all over predictors? Taking into account each other? Table 4 is unclear on whether this was all one model or separate models and whether dummy variables were constructed and all in one model?
All variables were entered into in one model (general linear model). The dependent variable was maternal functioning and the independent variables were mental health and socio-demographic variables that had relationship with maternal functioning with p<0.05. The variables of total score of mental health, mother’s age and husband’s age were quantitative variables and entered into the model as covariate and the others socio-demographic variables were qualitative and categorical that putted in fixed factors (fixed factors are categorical independent variables). In the general linear model, it isn’t necessary to construct dummy variables. SPSS does that by default. The last category of a qualitative variable is considered as reference category. We have revised this section.
Discussion
This discussion spends far too much time summarizing results and not enough space synthesizing and discussing implications. The discussion uses too much causal language and needs to be tempered quite a bit to more accurately reflect the cross-sectional nature of this work. It too strong to state things like “higher levels of maternal mental health positively impact overall functional status…” Similarly, this evidence is not strong enough to support some of the suggestions the authors. This work is an important step, but future prospective, longitudinal research is also needed. This needs to be stated and the suggestions need to be framed in light of the need for more research.
Thank you for your time investment in our work. We have modified and qualified our language regarding the study results and suggested prospective, longitudinal studies for the future..
- The conclusion needs expansion and tempering. What are potential next steps? What further research is need to then be able to inform on screening and prevention. Overall, though, it’s hard to comment in detail on the discussion as the results are so unclear. Once the results can be clarified, it will be easier to determine whether this discussion is appropriate.
- We have expanded the conclusion section. As per the above comments, the language around the results has been modified.
Reviewer 2 Report
The authors examined the relationship between maternal functioning and mental health in the postpartum period. They recruited 530 new mothers and assessed their parental functioning, using the Barkin Index of Maternal Functioning (BIMF) and mental health using the Mental Health Inventory (MHI). They found that the mother's mental health and its sub-domains had positive associations with infant care, mother-child interaction, psychological well-being, social support, management, adjustment, self-care, and maternal functioning. Also, higher total
mental health score, moderate-income, and receiving support for infant care predicted maternal functioning.
This is a well-written paper, yet the Limitation section needs to be elaborated. First, this study is based only on self-report measures, and there is no integration of self-report and observational mother-infant interaction measures to evaluate maternal functioning. Second, the authors excluded from their sample mothers who had experienced recent stressful events in their lives (a choice I am not fully understand, explaining the rationale for this choice would be helpful), and/or had a history of mental disorder (what was the cut-off criteria for “history of mental illness”, diagnosed by a doctor/psychiatric? ) which do not represent the circumstances of all new mothers.
Author Response
Reviewer 2
Comments and Suggestions for Authors
The authors examined the relationship between maternal functioning and mental health in the postpartum period. They recruited 530 new mothers and assessed their parental functioning, using the Barkin Index of Maternal Functioning (BIMF) and mental health using the Mental Health Inventory (MHI). They found that the mother's mental health and its sub-domains had positive associations with infant care, mother-child interaction, psychological well-being, social support, management, adjustment, self-care, and maternal functioning. Also, higher total mental health score, moderate-income, and receiving support for infant care predicted maternal functioning.
This is a well-written paper, yet the Limitation section needs to be elaborated. First, this study is based only on self-report measures, and there is no integration of self-report and observational mother-infant interaction measures to evaluate maternal functioning. Second, the authors excluded from their sample mothers who had experienced recent stressful events in their lives (a choice I am not fully understand, explaining the rationale for this choice would be helpful), and/or had a history of mental disorder (what was the cut-off criteria for “history of mental illness”, diagnosed by a doctor/psychiatric? ) which do not represent the circumstances of all new mothers.
Your comments are insightful and greatly appreciated. We have elaborated on the limitations section. We have also explained the reason for excluding mothers who had experienced recent stressful events in their lives and/or had a history of mental disorder in the Study Design and Participants subsection of Methods section.
Thanks
